# A sulfotransferase dosage-dependently regulates mouthpart polyphenism in the nematode *Pristionchus pacificus*

Linh T. Bui [1], Nicholas A. Ivers[1] & Erik J. Ragsdale [1]

Polyphenism, the extreme form of developmental plasticity, is the ability of a genotype to produce discrete morphologies matched to alternative environments. Because polyphenism is likely to be under switch-like molecular control, a comparative genetic approach could reveal the molecular targets of plasticity evolution. Here we report that the lineage-specific sulfotransferase SEUD-1, which responds to environmental cues, dosage-dependently regulates polyphenism of mouthparts in the nematode *Pristionchus pacificus*. SEUD-1 is expressed in cells producing dimorphic morphologies, thereby integrating an intercellular signalling mechanism at its ultimate target. Additionally, multiple alterations of *seud-1* support it as a potential target for plasticity evolution. First, a recent duplication of *seud-1* in a sister species reveals a direct correlation between genomic dosage and polyphenism threshold. Second, inbreeding to produce divergent polyphenism thresholds resulted in changes in transcriptional dosage of *seud-1*. Our study thus offers a genetic explanation for how plastic responses evolve.

[1] Department of Biology, Indiana University, 915 E. 3rd St., Bloomington, IN 47405, USA. These authors contributed equally: Linh T. Bui, Nicholas A. Ivers. Correspondence and requests for materials should be addressed to E.J.R. (email: ragsdale@indiana.edu)

Developmental plasticity, the ability to produce multiple phenotypes from a single genotype, is a ubiquitous feature of multicellular life. At its extreme, plasticity takes the form of polyphenism, by which discrete alternative morphs develop, often with disparate ecological roles and sometimes exhibiting evolutionary novelties[1,2]. Because its binary outputs imply a limited number of factors needed for a switch mechanism, polyphenism offers a tractable model for studying how plasticity is controlled and responds to selective pressures. Genetic regulators of morphological polyphenism have indeed begun to be revealed[3–6], and the selectable variation shown for polyphenisms[7,8] suggests that genetic changes in the control of polyphenism may ultimately be identified. However, the genetic architecture of polyphenism is still not well understood, and it has remained elusive how such architecture should change across species with different plastic responses under different ecological pressures.

The omnivorous nematode *Pristionchus pacificus* has a polyphenism in its adult mouthparts, allowing individuals to take different diets in response to local environmental cues experienced as postembryonic larvae (Fig. 1). Specifically, the nematodes develop either into a "stenostomatous" (St) morph, which has a narrow mouth specialised for feeding on microbes, or into a "eurystomatous" (Eu) morph with opposing, moveable teeth that allow a broader diet, including predation on other nematodes[9]. Production of the Eu morph, which confers a fitness benefit over the St morph when deprived of bacteria and offered nematodes as prey[9], is influenced by the local availability of microbial food as sensed by starvation[10] and local crowding by conspecifics[11]. How responsive (i.e., plastic) the polyphenism decision is to environmental changes varies across populations and species[3,12]—for example, the laboratory reference strain of *P. pacificus* is mostly (90%) Eu even when well-fed under laboratory conditions—such that the set point of the polyphenism threshold can vary, potentially offering more or less sensitivity to given cues according to local environmental pressures. At a genetic level, this polyphenism is regulated by the lineage-specific, dosage-dependent sulfatase EUD-1[3]. This completely penetrant regulator channels pheromone signalling[13], endocrine (DAF-12–dafachronic acid) signalling[10], and information from chromatin modifiers and antisense RNAs[14] into a single switch. EUD-1 activity depends on the nuclear receptor NHR-40, which was proposed to be at the transcriptional terminus of the switch mechanism[15]. Yet despite an emerging genetic knowledge of this polyphenism and of plasticity in general, how genetic changes result in the evolution of plastic responses is little understood. Here, through the identification of a new polyphenism switch-gene, we show that a plastic response (i.e., morph ratio) is decided by the relative dosage of putative signal-modifying enzymes. Specifically, we have discovered a sulfotransferase with dosage-dependent epistasis over EUD-1 to set a threshold for a plasticity switch. Furthermore, we show that environmental, laboratory-selected, and interspecies changes in plasticity correlate with changes in the relative dosage of these genes. Our findings thus provide genetic insight into how the molecular regulation of polyphenism evolves to produce new plastic responses.

## Results

### Several recessive mutants suppress a polyphenism switch gene.
As an unbiased approach to identify genes making up a polyphenism switch, we conducted a forward genetic screen for recessive suppressors of the mutant strain *eud-1(tu445)*, which is completely St (i.e., Eud, eurystomatous-form-defective). We mutagenised *eud-1(tu445)* mothers, isolated their putatively

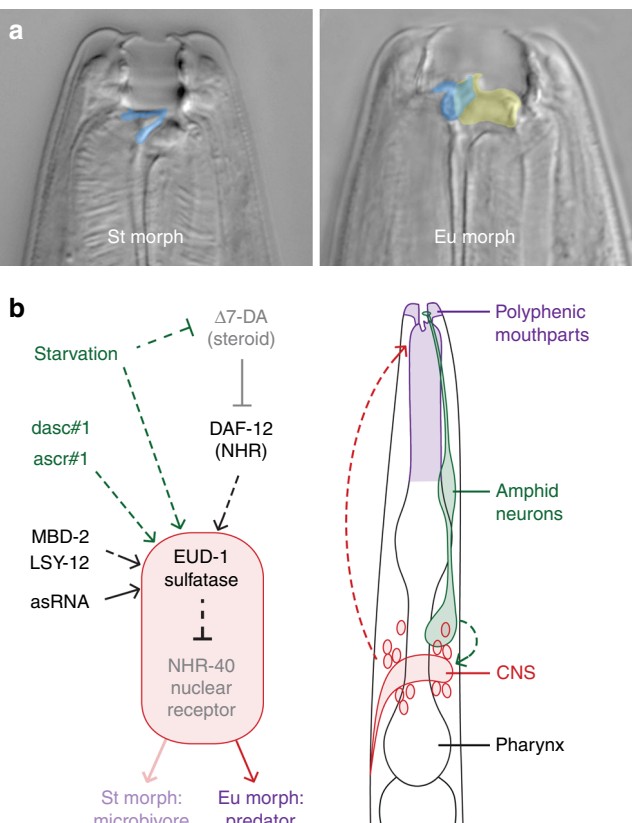

**Fig. 1** Developmental regulation of polyphenism in *Pristionchus pacificus*. **a** Alternative morphs (eurystomatous, "Eu," and stenostomatous, "St") have mouthparts that allow different ecological roles: the Eu morph differs in the shape of its dorsal tooth (false-coloured blue) and the presence of an additional, opposing tooth (yellow), which enable predatory feeding on other nematodes. **b** Known genetic factors regulating the mouth-form polyphenism (left) and their known or presumed spatial context in the developing larva (right). Sensitivity to environmental cues (green) of nutritional stress and crowding (pheromones dasc#1 and ascr#1) occurs during postembryonic development, where "starvation" (experimental withholding of food) has an influence as early as the first post-hatch (J2) stage[10] and the presence of other nematodes as late as the J3 and possibly J4 (pre-adult) stage[11]. Downstream of these cues, which in *C. elegans* are primarily intercepted and interpreted by several amphid neurons[55, 56], parallel regulatory elements including endocrine (dafachronic acid) signalling, chromatin modification (by LSY-12 and MBD-2), and *eud-1* asRNAs converge on a polyphenism "switch." This switch includes EUD-1 and NHR-40, which together comprise a signalling system that originates in the central nervous system (red) and ultimately decides between alternative feeding morphologies (purple). The decision is irreversible by the J4-adult moult, resulting in adult phenotypes best matched to the environment experienced as larvae

heterozygous $F_1$, and screened segregant $F_2$ for mutants showing an Eu phenotype. From a screen of ~10,300 haploid genomes, we identified seven recessive mutants. These mutants fell into one of three complementation groups, suggesting that the polyphenism switch comprises, and can feasibly be described from, a finite number of factors with non-deleterious mutant phenotypes. One of the complementation groups consisted of two mutant alleles, *sup(iub7)* and *sup(iub8)*, which were fully penetrant for their constitutive suppression of the Eud phenotype, even under well-fed conditions (Fig. 2a; Supplementary Table 1). We named the gene represented by these alleles *seud-1* (suppressor-of-*eud-1*).

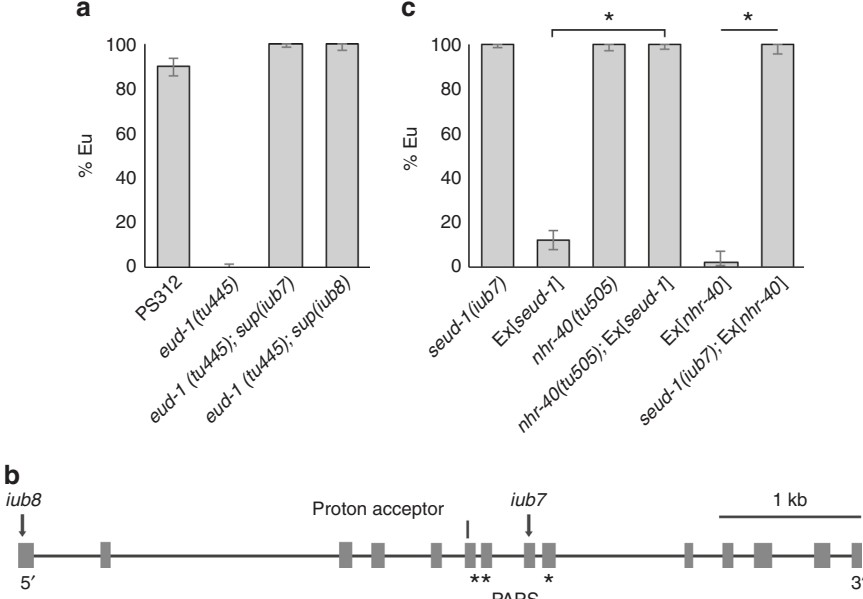

**Fig. 2** Mutants, gene structure, and epistasis of the plasticity regulator *seud-1*. **a** In contrast to the wild-type and *eud-1* phenotypes, mutants *iub7* and *iub8* are completely penetrant for the suppressor-of-Eud (and all-eurystomatous) phenotype. **b** Mutants uncovered were the null alleles *iub7* and *iub8*, both of which lack an incomplete sulfotransferase domain (catalytic proton acceptor + PAPS sulfate donor). **c** Mutant rescue of *seud-1* mutants with a transgenic construct, Ex[*seud-1*], indicates the dosage-dependent activity of this gene, whose function is reciprocally necessary with *nhr-40* for the polyphenism switch. Phenotype is the proportion of eurystomatous (Eu) individuals produced per clone. Scale bar, 20 μm. % Eu, proportion of eurystomatous nematodes. *$P < 0.0001$ ($\chi^2$-test). Whiskers represent a 95% confidence interval

**A sulfatase is suppressed by a sulfotransferase**. To identify the locus of *seud-1(iub7)* and *seud-1(iub8)*, we backcrossed each mutant and resequenced its genome. We scanned all annotated genes for non-identical and potentially harmful (nonsense, missense, and splice-site) mutations in both mutants. The entire genome yielded only one predicted gene, Contig20-aug8366.t1, with harmful lesions unique to each mutant. This gene, located on Chromosome I, encodes a cytosolic sulfotransferase and is one of several *P. pacificus* homologues of *Caenorhabditis elegans ssu-1*. In *C. elegans*, SSU-1 functions in dauer diapause, another type of environmentally conditioned development, acting with the steroid hormone receptor DAF-12/VDR in ASJ amphid neurons to suppress formation of dauer downstream of insulin (DAF-2) signalling[16,17]. Mutant lesions in *iub7* and *iub8* were nonsense mutations in Exon 8 and Exon 1, respectively, of the gene as annotated from expressed RNAs (Fig. 2b). The *iub8* lesion lies upstream of sequences encoding a proton acceptor (H[164]) for a substrate-binding domain, and *iub7* lies within a sulfate-donor (3′-phosphoadenosine-5′-phosphosulfate, PAPS) domain (Supplementary Table 2). Sequences for both functional domains are highly conserved across nematodes and the mammalian homologue SULT2A1 (Supplementary Fig. 1), suggesting the conserved function of SEUD-1 as a cytosolic sulfotransferase in *P. pacificus*.

**SEUD-1 is a genetic intermediary between EUD-1 and NHR-40**. Given the likely function of SSU-1 as a sulfotransferase, we hypothesised that this enzyme may act, like the sulfatase EUD-1, on a substrate upstream of NHR-40. To test this idea, we investigated its epistasis with *nhr-40*, particularly using a line of the null mutant *seud-1(iub7)* that was outcrossed to restore wild-type alleles at *eud-1* (Supplementary Table 3). Because *nhr-40* and *seud-1* both promote the St form in wild-type animals, we performed epistasis tests by combining loss-of-function mutations in one gene with transgenic overexpression in the other. These tests specifically used an *nhr-40* mutant allele (*tu505*) that, like all

previously recovered *P. pacificus nhr-40* mutants, incurred a non-synonymous mutation that is fully penetrant for its Eu-promoting phenotype[15]. First, we transgenically over-expressed a wild-type (PS312) construct of *nhr-40* in a null *seud-1(iub7)* mutant. This *seud-1(iub7)*; Ex[*nhr-40*] line resulted in the all-Eu phenotype of *seud-1(iub7)* rather than the mostly-St phenotype of the over-expressing line Ex[*nhr-40*] ($P < 10^{-4}$; Fig. 2c; Supplementary Table 4), indicating that *seud-1* either acts downstream of NHR-40 or, alternatively, is necessary to sulfate a signal that ultimately activates NHR-40-mediated transcription. In the former model, *seud-1* over-expression should suppress the phenotype of *nhr-40* mutants, whereas in the latter model the reverse should be the case. To distinguish between these possibilities, we transgenically over-expressed a wild-type genomic clone of *seud-1*, including its presumptive 5′ and 3′ regulatory elements, in *nhr-40(tu505)* mutants (Supplementary Table 5). Although transgenic Ex[*seud-1*] animals wild-type for *nhr-40* were mostly St, Ex[*seud-1*] animals with loss-of-function *nhr-40* alleles exhibited the *nhr-40* mutant phenotype, rather than the *seud-1* over-expression phenotype ($P < 10^{-4}$; Fig. 2c; Supplementary Table 4). Therefore, reciprocal epistasis tests indicate that SEUD-1 acts between EUD-1 and NHR-40 and has an enzymatic function necessary for NHR-40 function.

**SEUD-1 is expressed at the site of the polyphenism**. Because expression of *eud-1* and *nhr-40* does not overlap, and because *nhr-40* is pleiotropically expressed across several tissues[15], the anatomical scope of the polyphenism switch was previously unknown. We therefore studied the expression of *seud-1* to reconstruct the switch in space. To localise expression of *seud-1*, we generated a transcriptional, nucleus-localised, fluorescent (RFP) reporter using the same promoter sequence that was sufficient to over-express *seud-1*. This transgenic construct reported *seud-1* expression in the cells making up the anterior body wall (face), stoma (mouth), and anterior pharynx, which have

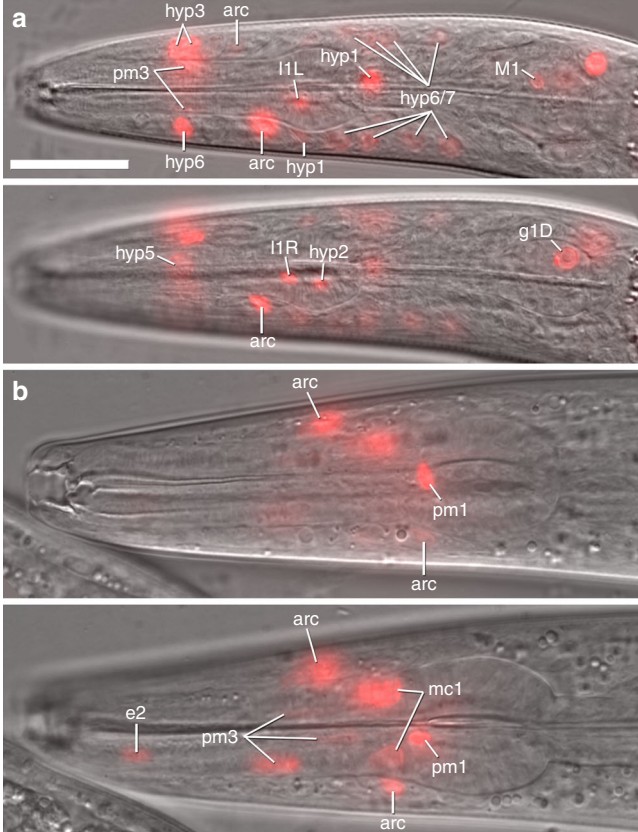

**Fig. 3** Spatial expression of *seud-1* in *Pristionchus pacificus*. Expression of a nucleus-localised *seud-1^promoter^::TurboRFP* reporter is shown for (**a**) a J3 larva and for (**b**) a stenostomatous adult hermaphrodite shortly after the final moult, each in two focal planes. J3 is shown in sagittal and left of sagittal planes; adult, in sagittal and right of sagittal planes. Expression is reported for pharyngeal myoepithelial (pm1, pm3, pm4), epithelial (e2, mc1), dorsal gland (g1D), tooth cell-innervating neurons (I1, M1), interfacial (posterior and anterior arcade, "arc"), and anterior epidermal (hyp) cells of, collectively, the anterior pharynx, stoma (mouth), and anterior body wall. Not all expressing cells, some of which (e.g., hyp, arcade cells) could not reliably assigned to a syncytium, are labelled. Expression ontogeny across life-stages suggests variation in timing among various components of dimorphic mouth morphologies, although expression consistently encompassed the breadth of morphology described or, in the case of pharyngeal neurons, presumed to be dimorphic in *P. pacificus*. Scale bar (for all images), 20 μm

homologues in *C. elegans*[18] and which together produce the feeding morphology that is dimorphic in *P. pacificus* (Fig. 3). We detected expression in these cells throughout postembryonic juvenile stages (J1–J4), dauer (a facultative, third-stage diapause larva), and young adults (Supplementary Fig. 2). Expression was consistently most pervasive across cell classes at the J2 and J3 stages (Fig. 3a), thereafter tapering in the J4 and early adult stages (Fig. 3b), and indeed at the adult stage, when morphs could be distinguished, no obvious qualitative differences in expression between Eu and St morphs were observed. Additionally, although individuals of a given life-stage varied in which particular cells were expressed when the animals were caught for examination, all larval stages were collectively shown to express *seud-1* in this set of cells. These findings suggest that this polyphenism switch factor is active in dimorphic tissue throughout postembryonic development, including diapause, until the polyphenism is expressed at the adult stage.

The specific cells expressing the reporter were those making up the anterior epidermal (hyp) syncytia, which together comprise the anterior body wall and secrete the outer lining of the mouth (cheilostom)[19], the region of the mouth and head that is dimorphic in width; the anterior and posterior arcade cells, which join the mouth opening to the pharynx[20]; epithelial cells e2 and mc1, and myoepithelial cells pm1, pm3, and pm4, which together comprise the pharyngeal muscle and secrete the extracellular teeth[21]. In addition to cells in the mouth and anterior pharynx, strong expression was detected at early juvenile (J2 and J3) stages in the pharyngeal gland (g1) cell, the nuclei of which are in the pharyngeal basal bulb (Fig. 3a). This cell comprises a gland whose duct terminates in the dorsal tooth, the secretions of which have been hypothesised to be involved in predatory feeding and which is, compared with *C. elegans*, highly innervated by the pharyngeal nervous system[18]. Other cell classes reporting *seud-1* were M1, I1, and possibly I2 (Fig. 3a), all of which innervate pm1, the cell actuating the dorsal tooth, and are thus hypothesised to reflect dimorphism in pharyngeal behaviour[18]. Because the cells reporting *seud-1* overlap with several cells in which *nhr-40* is expressed[15], *seud-1* and *nhr-40* likely control the polyphenism decision at the site of the morphological dimorphism. Together, these results indicate that *seud-1* is expressed at the terminal integration site of the polyphenism pathway in *P. pacificus*, whereas EUD-1 regulates signalling upstream of these target cells.

**Polyphenism is set by dosages of enzymes of opposed function**. Because transgenic over-expression of *seud-1* showed the gene to be dosage-dependent, we hypothesised that the relative dosages of *seud-1* and *eud-1* may together establish the threshold for the polyphenism switch. To test this idea, we manipulated wild-type copy number, a feature shown to affect expressivity in *eud-1* hemizygotes and heterozygous mutants[3]. Specifically, we used *eud-1* and *seud-1* mutants to generate all possible combinations of homozygotes, heterozygotes, and hemizygotes, from both cross directions, for both genes (Supplementary Table 6). As a result, relative copy number of functional *eud-1* and *seud-1* alleles was sufficient to gradate the ratio of morphs among the $F_1$ from all-Eu to all-St (Fig. 4a). All genotypes with one copy of each gene restored the phenotype of wild-type hermaphrodites (morphological females), which have two copies of each, indicating the relative dosage dependence of *eud-1* and *seud-1*. Moreover, animals with one copy of *seud-1* but two copies of *eud-1* were almost all-Eu, as the Eu-promoting *eud-1* is favoured, although a few animals were still St. Lastly, no number of copies of *eud-1* could rescue Eu formation in the absence of wild-type *seud-1*. Together, these results indicate that, under a given environmental regimen, relative dosage of the two genes set the threshold for the mouth-morph ratio of *P. pacificus*.

**Morph-inducing cues differentially regulate *seud-1***. Given the dependence of the morph ratio on relative genomic dosage of *eud-1* and *seud-1*, we investigated whether transcriptional dosage of these genes reflected differences in morph ratio induced under different environments. Namely, we predicted that the different morph-inducing cues would result in higher *eud-1*:*seud-1* transcript ratios in an Eu-inducing compared with a St-inducing environment. To test this prediction, we performed an analysis of RNA transcripts produced across multiple nutritive environments demonstrated to elicit different plastic responses in *P. pacificus*[22]. Because the only variable in this experiment was food source, we interpreted any differences in phenotype and gene expression to be the result of dietary cues. Results of this analysis showed that nematodes cultured on a relatively St-inducing diet of

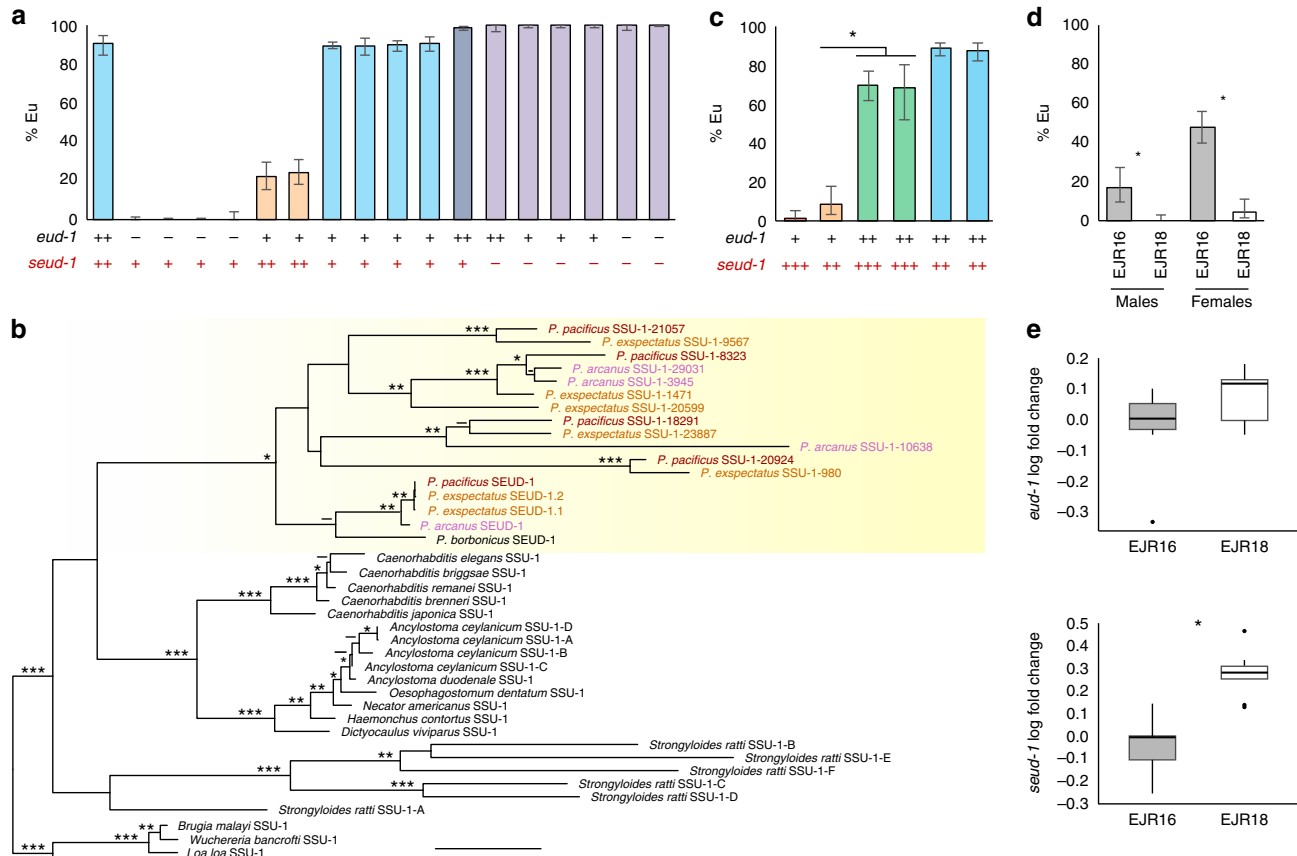

**Fig. 4** Genomic dosage, duplication, macroevolution, and inbred divergence of *seud-1*. **a** Compared to the wild-type (left bar), animals with the relative dosage of *eud-1* to *seud-1* manipulated (each dosage ratio marked by a unique bar colour) show a gradation of plasticity thresholds, as measured by the percentage of eurystomatous (Eu) individuals, produced in a common environment. Specific crosses used to produce dosages of *eud-1* to *seud-1*, as indicated along the *x*-axis, are given in Supplementary Table 6. Each plus-sign below chart represents a functional (wild-type) gene copy present in cross offspring; each minus-sign, a non-functional (mutant) copy. **b** *seud-1* (boldface font) is the product of gene duplications, particularly a large radiation of sulfotransferase-encoding *ssu-1* homologues, specific to a lineage including *Pristionchus* (yellow shading). Divergences even between three closely related species (marked in differently coloured fonts) indicates the dynamic divergence of these homologues, including the recent duplication of *seud-1* in *P. exspectatus*. Amino-acid sequences were analysed by maximum likelihood. Bootstrap support values: ***99–100%; **96–98%; *75–95%; −50–74%. **c** Manipulation of competing (*eud-1*:*seud-1*, each dosage ratio marked by a unique bar colour) gene dosages using hybrids with *P. exspectatus*, support the function of both duplicates and their consequences on the different plasticity phenotype observed for *P. exspectatus*. Specific crosses used to produce dosages of *eud-1* to *seud-1*, as indicated along the *x*-axis, are given in Supplementary Table 9. *$P < 0.0001$ ($\chi^2$-test). **d** Lines of *P. exspectatus* (EJR16, EJR18) were systematically inbred from a heterozygous parent strain to result in different polyphenism ratios. *$P < 10^{-4}$ ($\chi^2$-test). **e** Inbred lines of *P. exspectatus* diverged in transcriptional dosage (log₂ fold change of expression) of *seud-1* (i.e., *Pex-seud-1.1* + *Pex-seud-1.2*), in particular relative to *eud-1*, expression of which did not change ($P > 0.05$). Box plots show ΔΔCt values (centre line, median; box limits, upper and lower quartiles; whiskers, 1.5× interquartile range; points, outliers). *$P < 0.001$ (Tukey's test). Whiskers represent a 95% confidence interval

## Table 1 Environmentally induced expression changes in *seud-1*

| Food source | % Eu | *seud-1* log₂ FC | *p*-value | *eud-1* log₂ FC | *p*-value |
|---|---|---|---|---|---|
| OP50 | 89 | – | – | – | – |
| *C. curvatus* | 84 | 0.4338 | 0.186 | 0.7282 | 0.0269 |
| *C. albidus* | 50 | 2.1083 | 1.27E−12 | 0.7725 | 0.00250 |

Log fold change (log₂ FC) of *seud-1* and *eud-1* is shown for *Pristionchus pacificus* (strain PS312) grown under different morph-inducing environments (*Cryptococcus* yeast species offered as food) compared to a diet *E. coli* OP50. % Eu is the median proportion of the eurystomatous (Eu) morph out of total individuals screened, as previously reported on those diets[22]. *p*-values were obtained using an exact test with a Benjamini–Hochberg correction for multiple comparisons

*Cryptococcus albicus* yeasts resulted in higher enrichment of *seud-1* expression compared with a diet of *E. coli* OP50 (Table 1). In contrast, a relatively Eu-inducing diet of *C. curvatus* yeasts

showed no difference in either phenotype or *seud-1* expression. Further, *eud-1* transcript levels were lower on a diet of *C. albicus*, and only slightly lower in *C. curvatus*, further shifting the ratio of *eud-1* to *seud-1* transcripts on the former diet. Our results thus indicate that, in a single genetic background, different morph-inducing environmental cues result in relative transcriptional dosage of opposing factors making up the polyphenism switch.

**seud-1 homologues are dynamically radiating in Pristionchus.** To reconstruct the evolutionary origin of *seud-1*, we performed a phylogenetic analysis of *seud-1* and other identifiable homologues of *C. elegans ssu-1* in *Pristionchus* and outgroups (Supplementary Table 7). This analysis showed that three sibling species of *Pristionchus* each had at least four to seven homologues of *ssu-1*. Several of these genes have high inferred divergence and dubious orthology (Fig. 4b), despite the species' being close enough to interbreed[23]. In contrast, the most recent common ancestor of *Pristionchus* and outgroups likely carried a single *ssu-1*

homologue. Therefore, a radiation of *ssu-1* duplicates has been specific to a lineage including *Pristionchus* and coincides with the presence of polyphenism in evolutionary history[12]. A product of this radiation was *seud-1*, for which orthology could be established across *Pristionchus* species (Fig. 4b). However, even within this clade, a recent duplication of *seud-1* was detected, specifically in *P. exspectatus*. Pairwise counts of $d_N/d_S$ between the two *Pex-seud-1* duplicates (0.027), as well as between each copy and the reference allele for *P. pacificus* (0.012, 0.011), suggest that both are under strong purifying selection (i.e., $d_N/d_S \ll 1.0$) rather than harbouring an incipient pseudogene. Furthermore, the coding sequences of the two copies are >99% identical at the amino-acid level, suggesting a similar enzymatic function between duplicates.

**Gene duplication correlates with plasticity changes among species.** Because the *P. pacificus* polyphenism is sensitive to genomic dosage of switch genes, we hypothesised that the recent duplication of *seud-1* in *P. exspectatus* has changed the species' plasticity phenotype. The sampled strain of this species, prior to its laboratory inbreeding to an all-St phenotype, produced more St animals than most strains of *P. pacificus* under laboratory conditions[3]. In principle, duplication could provide an instant change in a plasticity response given the standing *trans*-regulation of factors making up a polyphenism switch: in *P. exspectatus*, this would result in the St morph being more likely to develop in a given environment. We tested whether *Pex-seud-1.1* and *Pex-seud-1.2*, which are both expressed (Supplementary Table 8), had a combined influence on the polyphenism decision, using *P. pacificus* mutants to create interspecific hybrids with different relative copy numbers of *eud-1* and *seud-1* (Supplementary Table 9). By examining F1 females from these crosses, we could standardise the regulatory genetic background of these genotypes, which has likely diverged in addition to gene dosage. As predicted, changing the ratio of copies of *eud-1* to *seud-1* (1:3, 1:2, 2:3, and 2:2) in hybrids resulted in a gradation of plasticity phenotypes (Fig. 4c). In particular, hybrid phenotypes showed a difference between double "heterozygotes" (1:2) and double "homozygotes" (2:3) from either *P. pacificus* mothers ($P < 0.0001$, Z-test) or *P. exspectatus* mothers ($P < 0.0001$, Z-test), consistent with both *Pex-seud-1* duplicates having an influence on the morph ratio in *P. exspectatus*. Taken together, these results show a clear correlation between the duplication of one of two opposing switch factors and a divergent plastic response (i.e., morph ratio) to a given environmental regimen.

**Inbred plasticity differences correlate with *seud-1* regulation.** Lastly, we examined whether changes in relative transcriptional dosage could be detected among haplotypes differing in their plastic responses. First, we quantified expression of *eud-1* and *seud-1* in three genetically distant *P. pacificus* isolates[24,25] that vary in their morph ratios in a standardised environment[3]. While two St-biased strains (RS5200B, RS5410), as predicted, showed lower ($P < 0.05$, Tukey's test) or a trend toward lower ($P = 0.1$, Tukey's test) *eud-1* expression than the California strain, *seud-1* expression was also lower in these St-biased strains ($P < 0.05$, Tukey's test; Supplementary Fig. 3), indicating that at least some variation must be attributed to regulatory differences beyond the simple scaling of *eud-1* and *seud-1* expression. Furthermore, no coding-sequence variation could be found to explain this phenotypic variation, perhaps due to additional differences in other polyphenism factors (e.g., NHR-40) or their target gene repertoire. Therefore, we next performed an experiment to test whether relative dosages could be bred for divergent phenotypes on a shorter time scale, specifically from a single, heterozygous laboratory population. To do this, we systematically inbred lines

of *P. exspectatus*, a gonochoristic species harbouring higher levels of heterozygosity than *P. pacificus*[25], possibly including for plasticity-regulating genes. Systematic inbreeding for 10 generations indeed resulted in lines being fixed for different polyphenism thresholds in both females ($P < 10^{-4}$, $\chi^2$-test) and males ($P < 10^{-4}$, $\chi^2$-test) in standardised environments (Fig. 4d). Furthermore, when we measured the relative expression of *eud-1* and total "*seud-1*" (*seud-1.1* + *seud-1.2*) in these lines, we found that an increase in St animals correlated with increased relative transcription ($P < 0.001$, Tukey's test) of combined *seud-1* duplicates (Fig. 4e). Thus, natural variation in transcriptional regulation could be selected to result in both in divergent plastic responses and in divergent switch-gene transcription, implicating these genes as potential targets for plasticity evolution.

## Discussion

In summary, we describe a genetic model for a signalling system regulating developmental plasticity, and we have identified molecular targets of plasticity evolution (Fig. 5). Our findings support two major insights about plasticity regulation and its capacity to change. First, we have detailed how a pair of enzymes can fine-tune a polyphenism threshold without requiring changes to signal production itself. Although hormones have been long known to mediate morphological polyphenism in animals[26], the molecular mechanisms that integrate polyphenism have been poorly understood. In *P. pacificus*, a plasticity threshold is mediated by opposing factors regulating local signalling function, particularly through sulfation and desulfation. Further, sulfation by SEUD-1 is instructive, promoting the activity of NHR-40, possibly by locally providing a metabolic precursor or modifying transport of an active ligand[27]. Alternatively, SEUD-1 and EUD-1 may regulate NHR-40 itself, such as through its acetylation

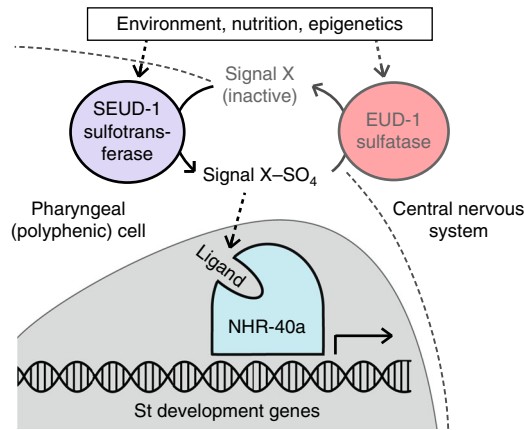

**Fig. 5** A genetic model for polyphenism regulation and its potential evolutionary targets. A signalling mechanism originating in somatic neurons is integrated at the site of the polyphenism, where *seud-1* and *nhr-40* are expressed. Results suggest a model whereby the maintenance of a sulfated signal, possibly modified extracellularly by the signal peptide-including isoform of EUD-1[3], promotes St-morph gene transcription (or, alternatively, Eu-morph gene repression) through NHR-40, particularly its complete (DNA-binding and ligand-binding) isoform (NHR-40a). Consistent with previous studies of morph-inducing cues, function, and expression of *seud-1* suggests that this switch mechanism may be responsive to cues throughout postembryonic development, before making its irreversible decision at the J4-adult moult. The threshold for the morph ratio is thus set by the relative dosage of two enzymes of opposing function, such that changes to this dosage can alter the threshold for the polyphenism under a given set of environmental conditions

state[28], although the apparent absence of EUD-1 from polyphenic tissue makes this scenario less likely. Signal modification would also be consistent with the inferred activity of *C. elegans ssu-1*, for which mutants show altered levels of sulfated sterols[14]. We therefore propose that a signalling molecule, still to be identified, is alternately modified to regulate its local activity at the targets of the polyphenism pathway.

Secondly, our study gives a genetic explanation for how genetic changes to a polyphenism mechanism produce new plasticity responses in different lineages. Opposing signalling modifiers offer an alternative evolutionary target besides the production of signals themselves, which are often pleiotropic and likely to be constrained[29,30]. This target also differs from a signal's ultimate receptor, to which mutations can have a dramatic impact on phenotype[31] and also be hindered by pleiotropy, as observed for NHR-40[32]. Thus, the balance between competing enzymatic regulators can be shifted to change the threshold response required for a switch. In principle, changes to such a system may be fine-tuned by allowing selection on opposing factors that individually channel separate, upstream influences on polyphenism. It is also possible that gene duplications, as observed in *P. exspectatus*, could result from gene amplification in such a system. In such a case, fixation of duplicates and the specialisation of new developmental regulators would thereafter be possible[33]. Although functional assays have revealed only limited polyphenism function for paralogues of *eud-1*[34,35], the more extensive radiation of *seud-1* homologues allows a better test of this idea. By revealing the precise factors controlling a polyphenism switch, it is now possible to reconstruct what changes to a switch have occurred across species that have diverged in their sensitivity to environmental pressures.

In conclusion, we have identified the genetic regulatory points that establish a threshold between plastic developmental phenotypes, and we have shown that modifications to these regulatory points correlate with divergent morph-ratios between species. Importantly, this model offers a foundation for identifying the genetic modifications that have occurred in other instances of polyphenism evolution, thereby revealing its generalisable features. The nematode family (Diplogastridae) that includes *Pristionchus* has many other species with this ancestral polyphenism, and among lineages there is a diversity of plasticity responses as well as plastic morphologies[12,36]. With our genetic description of a switch apparatus, it is possible to comparatively explore how evolutionary modifications to this switch—in regulation, function, or downstream targets—have facilitated a dramatic radiation of polyphenism-governed traits.

## Methods

**Nematode cultures**. All *Pristionchus* strains were maintained on 6-cm Petri plates containing nematode growth medium (NGM) agar seeded with 300 µl of *Escherichia coli* grown in L-broth. All cultures were maintained at 20–25 °C.

**Forward genetics screen**. Mutagenesis of the *P. pacificus* null mutant *eud-1 (tu445)* followed a described protcol[37]: self-fertilising hermaphrodites ($P_0$) were mutagenised at the J4 (pre-adult) stage by incubation in 47 mM ethyl methane-sulfonate at 20 °C for 3.5 h on a rocker at low speed (40 UPM), washed three times with 5 ml of M9 buffer, transferred onto individual 10-cm NGM plates, and allowed to produce ~20 offspring each. These $F_1$ self-fertilised to produce $F_2$ on the same plates, and once most $F_2$ had reached the adult stage, we screened for nematodes of the Eu morph, which suppressed the Eud (all-St) phenotype. Eu $F_2$ were cloned into individual culture lines. Phenotypes of >100 $F_3$ clones of each line were screened to confirm the mutant phenotype to be both homozygous and recessive. Pairwise complementation tests were performed for the seven isolated recessive mutants, and these tests grouped alleles *iub7* and *iub8*. These two mutants were then backcrossed to the *eud-1(tu445)* strain four and six times, respectively, using the following strategy: for two rounds of backcrossing, mutant hermaphrodites were first crossed to *eud-1(tu445)* males, after which $F_1$ were backcrossed to the strain *pdl-2(tu463); eud-1(tu445)*, a strain in which the parental line had been marked with a recessive "dumpy" mutation[15]; $BC_1F_1$ were then selfed to recover Eu $BC_1F_2$, which were cloned and then backcrossed further as above.

**Phenotype scoring**. Mouth phenotypes were scored as previously described[11]. In short, the Eu phenotype was determined by the presence of (i) a claw-shaped dorsal tooth, (ii) a large, hooked subventral tooth, and (iii) a mouth wider than deep; the St phenotype was determined by (i) a narrow, dorsoventrally symmetrical dorsal tooth, (ii) absence of a subventral tooth, and (iii) a mouth narrower than deep. Rare (<0.5%) intermediates between the two morphs were excluded from phenotype counts. Phenotypes were scored using differential interference (DIC) microscopy using a Zeiss AxioScope, except in the $F_2$ screen for mutants, for which a Zeiss Discovery V.20 stereoscope was used for higher throughput. At least 60 individuals, with one exception ($n = 25$), were scored for each genotype (Supplementary Tables 4, 6, 9).

**Genomic resequencing for mutant identification**. To prepare samples for whole genomic sequencing, nemtatodes from five agar plates of each strain were washed with M9 buffer in a 50-ml conical tube. Nematodes were then washed with 0.9% NaCl with ampicillin (50 µg/ml) for 12 h, after which the nematode pellet was collected and washed with 0.9% NaCl with ampicillin (50 µg/ml) and chloramphenicol (25 µg/ml) for 12 h. After centrifuging at 1300×g for 4 min, the nematode pellet was collected and genomic DNA was extracted using a MasterPure DNA Purification Kit (Epicentre) and was quantified using an Agilent TapeStation 2200. DNA libraries for sequencing were prepared using a Nextera kit, diluted to a concentration of 0.45 pM in 0.1% EB-Tween, and pooled as 6-plex. The libraries were sequenced as 150-bp paired-end reads on an Illumina NextSeq 300 to a theoretical coverage of 24×. Reads from raw sequencing data were mapped to the *P. pacificus* genome (version Hybrid1, www.pristionchus.org) using the Burrows Wheeler Alignment tool[38], and variants were called using SAMtools[39].

**Identification of mutant lesions**. With the lists of potential variants for *iub7* and *iub8*, we identified the causal lesions as follows. First, we excluded all variants from each list that were shared with the resequenced genome of the mutagenised *eud-1 (tu445)* strain[3] and with the dumpy *pdl-2(tu463)* strain (voucher EJR1018) used to create the *pdl-2(tu463); eud-1(tu445)* strain for backcrossing. Variants in predicted coding sequences were categorised with respect to the Hybrid1 assembly and AUGUSTUS annotation for the genome of *P. pacificus* (www.pristionchus.org). From the intersecting lists of variants, we excluded SNPs manually determined to be artefactual using the Integrative Genomics Viewer[40]. Following the identification of the mutant locus as Contig20-aug8366.t1 from resequenced genomes, mutant lesions were verified by Sanger sequencing (Supplementary Table 3).

**Outcrossing of the *eud-1* mutation**. To determine the phenotype of a null *seud-1* mutant without a background *eud-1* mutation, we outcrossed the mutant *seud-1 (iub7); eud-1(iub8)* to the wild-type reference ("California," PS312) strain of *P. pacificus*. For the $P_0$ cross, we used males of the mutant strain and California hermaphrodites, so that $F_1$ males would inherit the autosomal *seud-1* mutation but not the X-linked *eud-1* mutation. $F_1$ males were then outcrossed again to California hermaphrodites, virgin ($BC_1$) offspring of which were let to self-cross to re-segregate mutant *seud-1* alleles from putative heterozygotes. Multiple $BC_1F_2$ lines were cloned and >200 individuals per line were screened to confirm their all-Eu phenotype. Finally, the presence of the identified *seud-1(iub7)* lesion and the absence of the *eud-1(tu445)* lesion were additionally confirmed by PCR and Sanger sequencing (Supplementary Table 3).

**Over-expression of *seud-1* by genetic transformation**. Transgenic *P. pacificus* nematodes were generated by microinjection of the gonads (germline) of adult hermaphrodites, which were then selfed, following which offspring with a reporter gene were screened among the $F_1$[41]. To rescue function of and over-express *seud-1*, ovaries of outcrossed *seud-1(iub7)* mutants were injected with an 11-kb genomic clone containing the putative 3.9-kb promoter region of wild-type *seud-1* and a 6.7-kb coding sequence including its 330-bp 3′ regulatory region (10 ng/µl). This construct was delivered with the co-injection marker *egl-20^promoter^::TurboRFP* (10 ng/µl) and digested genomic carrier DNA (60 ng/µl) from the recipient strain. To over-express *nhr-40* in *seud-1* mutants, an extrachromosomal array from a *nhr-40* rescue line generated previously[15] was crossed into the outcrossed *seud-1 (iub7)* strain.

**Crosses to study effects of relative gene dosage**. Genetic crossing experiments were performed to test for haploinsufficiency and opposing dosages between *eud-1* and *seud-1*. In our cross panel, all possible genotype combinations (i.e., homozygotes, heterozygotes, and hemizygotes) of wild-type (California) and mutant (*tu445* and *iub7*) alleles of *eud-1* and *seud-1* were created. The backcrossed, double-mutant line *seud-1(iub7); eud-1(tu445)* (voucher EJR1029), the outcrossed line *seud-1(iub7)* (voucher EJR1039), the backcrossed (×6) parental line *eud-1(tu445)*, and the California strain were used for creating these genotypes. To correctly identify $F_1$ instead of self-offspring from these crosses, all polyphenism mutants were marked with the recessive dumpy marker *pdl-2(tu463)*, such that cross-offspring could be easily distinguished by the rescue of a non-dumpy phenotype. In addition to using previously generated lines[15], we created the double mutant *seud-1 (iub7); pdl-2(tu463)* and triple mutant *seud-1(iub7); pdl-2(tu463); eud-1(tu445)*.

Homozygosity of mutations was confirmed by Sanger sequencing of all alleles in >10 clones per line (Supplementary Table 3).

**Localisation of seud-1 expression**. We localised expression of *seud-1* in *P. pacificus* using a transcriptional reporter. To express this reporter, we transformed *P. pacificus* germline cells along with co-injection markers as described above. Because *seud-1* is dosage-dependent, we predicted its expression in the California strain to be low, as shown previously for the St-promoting *nhr-40*[15]. Therefore, we used *P. pacificus* strain RS5200B for transformation, as this strain is a natural variant that is highly St in laboratory culture[3] and was thus more likely than the California strain to express *seud-1* at detectable levels. The construct used for genetic transformation was ligated from the following sequence fragments: (i) the 4.2-kb putative promoter region of *seud-1*, (ii) a coding sequence consisting of a start codon, a nuclear localisation signal, and *TurboRFP*, and (iii) the 3′ untranslated region of the gene *Ppa-rpl-23*. The construct was cloned into pCR4 TOPO vector and maintained in a TOP10 *E. coli* clone, which served as its source for microinjection experiments. Two independent transgenic lines were generated, and 10 animals at each life-stage (for adults, 5 St and 5 Eu) were examined using a Zeiss AxioImager.

**Environmentally altered eud-1 and seud-1 expression**. Raw RNA reads, which were produced by nematodes raised on *E. coli* OP50, *Cryptococcus albicans*, and *C. curvatus*[22], were downloaded from the NCBI database (accession number: SRP081198). The sequence published as UMM-S71-39.9-mRNA-1 was identified as *seud-1*, and UMM-S2877-2.55-mRNA-1 as *eud-1*. Paired-end reads (2 × 150 bp) from two biological replicates for each condition were aligned to the *P. pacificus* reference genome (version El Paco, www.pristionchus.org) using STAR (version 2.5.3)[42] under default settings. Read counts were achieved using the function FeatureCounts in Rsubread Bioconductor package (version 1.24.2)[43] and genes with a low number count (i.e., low-expressed) were filtered out. Reads mapped to *seud-1* and *eud-1* genes were examined manually in IGV viewer (version 2.3.77)[40] to eliminate false mapping. Differential expression analysis was performed in edgeR[44], with *p*-values adjusted according to the Benjamini–Hochberg procedure to restrict false discovery rate.

**Phylogenetic analysis**. Evolutionary history of *seud-1* homologues in *Pristionchus* and outgroups was inferred from predicted homologues mined from available nematode genomes using the criterion of reciprocal best BLAST similarity with *C. elegans ssu-1*. For reconstruction of the phylogenetic history of *seud-1*, predicted amino-acid sequences encoded by identified *ssu-1* homologues were aligned using the E-INS-I algorithm and default settings in MAFFT (version 7.1)[45]. The gene tree of predicted SSU-1 amino-acid sequences was inferred under the maximum likelihood (ML) criterion, as implemented in RAxML (version 8)[46]. Fifty independent analyses were performed. Analyses invoked a Whelan and Goldman model with a gamma-shaped distribution of rates across sites. Bootstrap support for the most likely tree among all runs was estimated by 500 pseudoreplicates.

**Selection analyses of seud-1 duplicates**. We tested for purifying selection among *seud-1* alleles using a pairwise counting method of $d_N/d_S$[47], as implemented in the programme PAML[48]. We performed three comparisons: (i) *Pex-seud-1.1* to *Pex-seud-1.2*; (ii) *Pex-seud-1.1* to *Ppa-seud-1(California)*; and (iii) *Pex-seud-1.2* to *Ppa-seud-1(California)*.

**Hybrid crosses to study effects of gene duplication**. Hybrid crosses followed the same logic described for studying relative gene dosage in *P. pacificus*, with the following modifications. In crosses of *P. pacificus* males to *P. exspectatus* females, a phenotypic (e.g., dumpy) marker was not used or needed, as *P. exspectatus* requires sexual reproduction to produce offspring. Also, only female hybrids were used to score phenotypes, as male F₁ from reciprocal crosses have different genetic backgrounds that were previously found to have asymmetrical epistatic effects on the mouth polyphenism[3].

**seud-1 and eud-1 expression in natural polyphenism variants**. To quantify transcripts from the California strain and two St-biased strains (RS5200B and RS5410), we used J3 and J4 nematodes from each of these populations. To collect nematodes mostly at these stages, cultures were synchronised by starting with eggs released by bleaching gravid mothers[49], 48 h after which nematodes were washed from plates and pooled. Following centrifugation of collected nematodes at 1300×g for 4 min, nematode pellets were resuspended in 1 ml of TRIzol. For RT-PCR, total RNA was extracted from the homogenised solutions of nematodes in TRIzol using the Zymo Direct-zol RNA Miniprep kit (Zymo Research). 1 μg of total RNA was used for first-strand cDNA synthesis with random hexamers in a 20-μl reaction using the SuperScript III Reverse Transcriptase kit (Invitrogen). cDNA was diluted five times, and 1.5 μl of the final volume was used for a 10 μl PCR. Real-time PCR to quantify expression was performed on a Roche LightCycler 96 system, using SYBR GREEN reaction mix and the manufacturer's (Roche) software and with the genes encoding beta-tubulin (*tbb-4*) and an iron-binding protein (Y45F10D) as the reference genes[50]. Primers for RT-qPCR of *P. pacificus seud-1* and *eud-1* are listed in Supplementary Table 10. The PCR cycle was: 10 min at 95 °C, followed by 45

cycles of 10 s at 95 °C, 20 s at 52 °C, and 15 s at 72 °C, with a single fluorescence read at the end of each extension. Each PCR reaction was performed on three independent biological replicates and three technical replicates for each group. Melting-curve analysis and gel electrophoresis were performed to ensure the absence of non-specific products or primer dimerisation, and PCR efficiency was identified with a 5-log titration of pooled cDNA. Relative expression levels were determined using the ΔΔCt method[50], with California strain designated as the control group.

**Genetic inbreeding of P. exspectatus polyphenism variants**. For systematic inbreeding to fix different plasticity phenotypes, we started with a source population of *P. exspectatus* (RS5522) previously shown to harbour heterozygosity in genes affecting the mouth-morph ratio[3]. This population was recovered from a frozen voucher of a culture that, prior to freezing, had been established from multiple individuals collected from a vector beetle[23] and had been kept in laboratory culture for about 1 year on large-diameter (13 cm) plates, with subcultures established through "chunking"[51], whereby dozens of individuals were transferred. This voucher comprised nematodes from ~10 separate cultures maintained under the conditions above and frozen according to standard protocols[37]. This population was the parent source of the inbred strain RS5522B, in which a Eud phenotype had previously been fixed through inbreeding[3]. From the revived RS5522 strain, we systematically inbred two new lines for 10 generations each, using a single male–female pair at each generation. Because systematic inbreeding of a dioecious, or obligately outcrossing, species was likely to lead to negative fitness effects on the populations, 10 individual pairs were mated at each generation, with the subsequent generation being established from the most fecund pair; furthermore, all pairs from the previous four generations were kept stable at low temperature (6 °C) as insurance in case lines from selected pairs suffered inbreeding generations later. All crosses were performed under standardised environmental conditions as described for experimental crosses above. As a result of this inbreeding method, we were able to establish lines, which were in turn frozen as vouchers that were both inbred and without obvious negative effects on brood size and generation time. Additionally, because the thawed voucher was highly St-biased, we promoted the retention of the polyphenism (i.e., as opposed to a fixed Eud phenotype), by selecting for the Eu morph for two generations. Specifically, males and females selected for breeding during the first two generations were both Eu, whereas in subsequent generations inbreeding was random with respect to mouth phenotype. The variable morph-ratio phenotypes of the resulting inbred lines (EJR16 and EJR18) were then recorded under a standardised environment as described above.

**seud-1 and eud-1 expression in inbred polyphenism variants**. To quantify transcripts from inbred *P. exspectatus* lines, mixed-stage nematodes were collected and prepared for quantitative real-time PCR (RT-qPCR) as described above for expression quantification in natural polyphenism variants. Because the coding sequences of *Pex-seud-1.1* and *Pex-seud-1.2* are highly similar, primers matching both genes were used to amplify the total number of *Pex-seud-1* transcripts. Two non-overlapping pairs of primers were used, and transcripts of both *Pex-seud-1.1* and *Pex-seud-1.2* were confirmed by manual inspection of trace files of sequences for both RT-qPCR amplicons. RT-qPCR followed the procedure described above for the quantification of *P. pacificus* variants, with *tbb-4* (encoding beta-tubulin) used as the reference gene[50]. Relative expression levels were determined with line EJR16 designated as the control group. To quantify *seud-1* expression, two primer sets were used and all data from two separate runs with the set "*seud-1A*" (Supplementary Table 8) and one run with the set "*seud-1B*" were combined. PCR products were purified and submitted to Sanger Sequencing (Eurofins Genomics LLC) to confirm the presence of both duplicate genes. *eud-1* expression was quantified from two separate runs using *eud-1* primers (Supplementary Table 8).

**Statistical analyses**. All statistical analyses were conducted in R (version 3.4.3)[52]. Polyphenism phenotypes were recorded as proportional data, specifically the percentage of Eu per total number of individuals screened. To determine whether there were differences in morph ratios, we used generalised linear models with binomial error and a logit link function, designating genotype as the explanatory variable. Significance was assessed for (i) epistatic interactions of *nhr-40* and *seud-1* (Fig. 2a); (ii) genotypes from hybrid crosses; (iii) differences between *P. exspectatus* inbred lines (Fig. 4d). Replicates within lines were included in each model, and significance of differences was determined using a χ²-test for pairwise comparisons, and a *Z*-test where multiple comparisons were made. Significant differences between specific contrasts (i.e., between results of hybrid crosses) in the final models were determined using post-hoc Tukey's honest significant difference tests as implemented in the package lsmeans[53]. For graphical representation of phenotype data, mouth-morph frequencies were pooled across replicates within lines, with 95% confidence intervals estimated using a binomial test; these results are qualitatively identical to those derived from the generalised linear model.

Significance of differences in expression of *eud-1* and *seud-1* transcripts among wild isolates of *P. pacificus* and between inbred *P. exspectatus* lines was determined by a linear mixed model, which accounted for repeated measures within biological replicates, using the lme function in the nlme package[54]. Log fold change was used

as the response variable, strain as a fixed explanatory variable, and biological replicate as a random variable. For tests involving *P. exspectatus*, this method also confirmed that there were no differences among experimental runs (i.e., different primer pairs) for quantification of *seud-1* expression.

## Data availability

Gene sequences original to this study (*P. arcanus*) and raw phenotypic data have been deposited in the Figshare repository under https://figshare.com/articles/Bui_Ivers_Ragsdale-deposit_xlsx/6708674.

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

## Acknowledgements

We thank the Indiana University Center for Genomics and Bioinformatics for DNA library preparation and genomic resequencing, Susan Feldt and Ryan Mueller for their help with nematode crosses, R. Taylor Raborn for suggestions on transcriptomic analyses, and Meagan Pritchard for media preparation and strain keeping. We also thank Ralf Sommer and Christian Rödelsperger (Max Planck Institute for Developmental Biology) for sharing their unpublished genome sequence of *P. arcanus*. This work was funded by the National Science Foundation (IOS 1557873).

## Author contributions

E.J.R. conceived and designed the study; L.T.B. and N.A.I. conducted the experiments; L.T.B., N.A.I., and E.J.R. analysed the data; L.T.B., N.A.I., and E.J.R. wrote the manuscript.

## Additional information

**Competing interests:** The authors declare no competing interests.

