## [Peer Review File · Nature Communications]

Reviewers' Comments:

Reviewer #1:

Remarks to the Author:

This is in general a well written and clean manuscript that provides evidence for components of a working model for a switch mechanism underlying an example of developmental plasticity or polyphenism (two forms) in nematode mouthparts; their results also indicate something about how the switch mechanism can change in the evolutionary diversification of related nematode lineages. More data will be required to open up the switch mechanism and its evolution more completely but I believe the work presented here is novel and worth publishing at this stage.

Overall although I am not able to judge the full details of the molecular genetics used in this paper, I consider the overall approach to be valid and the results to be of some wider interest to those interested in how polyphenisms and developmental plasticity is linked to evolutionary diversification, and how it may evolve in connection with molecular switch modules.

I do think the Figures could be smartened up and in some cases designed to more effectively cover how the plasticity works and functions. Some issues here, or at least questions raised are:

At which developmental stage are the nematodes sensitive to their environment – and especially to the presence/absence of potential prey? Also, how does this sensitivity relate to the pathways described in the figures and in particular to the working of the switch mechanism suggested in Fig 4. Why are there three dashed 'epigenetic' arrows in Fig 4 but only one for environment? Does this environmental influence of diet availability (or indeed those for epigenetics) coincide with the initiation or development of the pharyngeal morphology (apparently around J4-adult molt, Fig 2)? I note that in many other systems of polyphenism in insects the environmental cue can by some time precede hormone signals that in turn regulate differing expressions of developmental genes and pathways. Thus, even if details are elsewhere in the literature, I would like to see more clarity of the roles of 'starvation' and 'Ascarosides' at the top of Fig 1, as well as more clarity where possible (or if not possible some discussion thereof) of the top slice of Fig 4. Personally, I would be more cautious about the 'optimizing morphology to the environment' phrase in Fig 1; matching or association perhaps but optimizing – have the necessary experiments been done, including in natural environments? In this line of thinking, the phrase at the start of para 2 in the introduction '... to take different diets in response to local environmental cues' could I think be rewritten to make it clearer what is meant.

I would value some more information on sample sizes – for example in Fig 2 part C, how many individuals were assayed, how much variability was seen and also which stages of development were surveyed (similarly for S2; and text section: SEUD-1 is expressed at the site of the polyphenism)? Otherwise, I find the phrase 'thus controlling the switch at the site of plasticity' to be more open to doubt than the text in Fig legend suggests (again cf text para). This also goes back to uncertainty at least in this MS about the timings of the environmental sensitivity, the switch mechanism and the development itself.

In the Section headed Genetic inbreeding ..., the word 'presumably' is a red flag – what does it mean, how large was the voucher sample, what was limited and what were the culture conditions? I am not a nematode person, and so would like to know what are the likely fitness effects of ten generations of extreme (single pair) inbreeding on fitness; this seems extreme and would in the insects I work with a) be impossible and b) produce all sorts of negative effects on fitness and phenotype. What were the conditions of so-called artificial selection (for only two generations) and from what is detailed here I have no idea of how it was performed (e.g. intensity of selection, numbers used, assays performed and in which environmental conditions)? This relates, for example, to interpreting the final sentence of the Results. Perhaps some more details in a section in the SI.

In the introduction it is not clear to me what exactly the phrase 'we identify and alternative

polyphenism switch gene' refers to given what goes before.

I think adding some of the details indicated above would a) make the manuscript more readable and more likely to be used by non molecular-based researchers of developmental plasticity, and b) improve the description of the wider context of the study.

Reviewer #2:

Remarks to the Author:

By means of mutagenesis suppressor screen, this study has identified the sulfotransferase *seud-1* gene as a suppressor of *eud-1*, known to regulate the mouth polyphenism of *P. pacificus* – an intriguing forming of developmental plasticity associated with different feeding modes/substrates. In contrast to previously uncovered genes affecting the production of these alternative mouth morphs, *seud-1* is shown to be expressed in pharyngeal cells, thus potentially representing a gene involved in the terminal specification of feeding morphologies. The study also presents a number of genetic experiments, which indicate that *seud-1* acts in a dosage-dependent fashion, and further “competes” with *eud-1* expression to modulate the relative production of the two alternative phenotypes. In addition, the study has examined the evolution of *seud-1* and its orthologues in related *Pristionchus* species, in particular, in *P. exspectatus*, possessing *seud-1* duplicates. The study thus provides new insights into the genetic regulation of a polyphenism, and it is suggested that gene dosage amplification could play a particular role in the evolution of this polyphenism, or perhaps plasticity in general.

Understanding the molecular regulators of plasticity/polyphenisms and their evolution is a key aim of current evolutionary and developmental biology. The study results are therefore very valuable and of broad interest; specifically, the advance our understanding of an important emerging study system of plasticity, the *Pristionchus* feeding morph polyphenism. While the discovery of *seud-1* and its analysis are very interesting and of high relevance, I have two main criticisms: First, the manuscript, data presentation and interpretation are often difficult to follow, and the writing is frequently vague or imprecise. Second, some of the experimental data are rather weak in supporting the conclusions drawn, in particular, concerning the evolution of *seud-1* and its link to the feeding morph polyphenism; a number of obvious, relatively simple experiments could be carried out to corroborate certain conclusions.

I detail these two main points and additional comments below.

Starting with the abstract and introduction, I encountered many unclear statements and sentences regarding the study system. Often, generic terms, like plasticity or “plasticity phenotype” are used without making explicit reference to the phenotype studied (e.g. production of a specific morph type). The use of such generic terms introduces ambiguousness and confusion. For example: The statement “how genetic changes result in the evolution of novel plasticity phenotypes remain unknown” is unclear. What is a novel plasticity phenotype of interest here? Reaction norms? Morphology of alternative phenotypes? Similarly, the statement “Furthermore, we show that plasticity, by being sensitive to gene dosage, can evolve directly by gene amplification, in addition to expected stepwise transcriptional changes” is not explicit about what plasticity refers to.

For readers not very familiar with the study system, I would guess there is some confusion about the definition of plasticity and environmental cues regulating the induction of phenotypes. In classical cases of plasticity/polyphenism, there are specific cues inducing alternative phenotypes; however, in the case studied here, alternative phenotypes are generated even under a constant environment (e.g. bacteria only). The relative production of the alternative phenotypes is thus not tightly linked to the presence of alternatives cues – perhaps this should be made clear; moreover, none of the experimental work involves any modulation of environmental cues. It therefore

remains unclear, how environmental variation connects to the production of alternative feeding morphs and how this relates to the genetics studied. I would therefore recommend explaining the study system better in the introduction, in particular, to elaborate on how morph allocation occurs in constant environment, and why natural isolates produce 90% Eu morphs in bacterial environments.

RESULTS / DATA

- Data on morph production of recovered mutants is not quantitatively shown – what are their effect sizes, expressivities? Are they sensitive to environmental variation?
- Expression analysis using the reporter strain is minimalistic - what is the time course of expression? How does expression differ between different morphs? Can environmental cues modulate *seud-1* expression?
- Epistasis analysis is primarily based on over-expression lines – are there any relevant null alleles?
- There is no analysis of potential variation in *seud-1* (sequence/expression, e.g. of transcripts) across different *P. pacificus* wild isolates and connection to production of alternative phenotypes. This would be rather obvious and relevant to do.
- Genetics: The genetic analyses presented in Figures 3A and 3C are somewhat difficult to follow (the labelling of crosses could be simplified for figures). This is further complicated by the rather non-standard genetic terms used to describe results of crosses, and subsequent interpretations. For example, I don't see the need to talk about "competition" among the two enzymes and the sentence "Because a plasticity is set by the competition of two opposing enzymes, selection for differential regulation or genomic amplification of these factors provides a binary "dial" for a plasticity to evolve" is unnecessarily difficult. In addition, despite some explanations in the materials and methods section, I didn't fully understand how statistics were performed (e.g. frequency comparisons pooling of replicate crosses?).
- Analysis of *seud-1* duplicates in *P. exspectatus*: I found the hybrid genetics involving the two species rather weak and indirect experimental evidence to make the argument that "Gene amplification results in new plasticity phenotypes across species". Why not simply knock out/down these genes in this species?
- The experiments on "Laboratory evolution of plasticity occurs through competing switch genes" involved derivation of two lines by some initial selection plus inbreeding are in my view not very conclusive. While suggestive, they do not causally demonstrate that difference in Eu morph production stems from measured *seud-1* differences.
- Given the above points, I think multiple conclusions/statements in the paper are premature, including the following:
 - "Abstract: Further, recent duplication and laboratory selection of *seud-1* in a sibling species has amplified the gene's control of the polyphenism's threshold, identifying this gene as a target for plasticity evolution."
 - "Taken together, these results show that opposing switch factors have been amplified and are experiencing regulatory divergence in this species, resulting in a new plasticity response to a given environmental regimen."
 - In conclusion, we have identified a genetic mechanism for establishing a threshold between plastic developmental phenotypes, and we have shown how – in the case of *P. exspectatus* – such a mechanism has been modified to produce a novel plasticity response.

- In addition, certain claims are somewhat exaggerated, e.g. in the abstract “Our findings thus provide the first genetic insight into how the molecular regulation of polyphenism evolves to produce new plasticity phenotypes”. This statement is incorrect – a number of examples in insect polyphenisms or *C. elegans* dauer formation are known.

Reviewer #3:

Remarks to the Author:

This paper tackles an interesting phenomenon in animal differentiation in which genetically identical members of a species acquire different morphological traits. This kind of effect is frequent in ant and insect species, and also occurs in the nematode *Pristionchus pacificus*, specifically in the context of feeding organ morphogenesis. The current work is introduced as an exploration of the mechanism by which this type of phenotypic variation, called polyphenism, occurs in *Pristionchus pacificus*. Previous work had identified a sulfatase and a nuclear receptor hormone, acting in that order, to regulate alternative developmental fates of mouth parts. Here, a third gene is characterized, *seud-1*, identified as a suppressor of sulfatase mutants, which all exhibit only one form of mouth development.

The authors characterize interactions with the other genes in the pathway, examine the expression pattern of the gene, and examine its variations in evolution with regards to copy numbers. They conclude that *seud-1* acts together with *eud-1* as influencing the ratios of the different mouth-part phenotypes in a given strain.

The paper is clearly written, and the experiments are appropriate. The use of a “non-standard” model system is exciting, although by this point *P. pacificus* has many tools available. I have two main issues that I think are important to address that may impact the significance of the findings:

1. The authors set up the paper as a study to understand the origin of polyphenism. However, they appear to fall short of this. The problem is that they look at mutants or isolates having increased or reduced expression of genes and show that phenotypic ratios can be altered. But that does not indicate that the gene involved is the actual site of regulation when phenotypic choices are made. All that could be happening here is that they are altering the set points for the system, but not the regulatory points. It seems to me that the only way to convincingly demonstrate that a gene is the site of phenotypic control, is to track its expression or activity in animals displaying the different phenotypes and show a 1-1 correlation. Even that may not be sufficient if it is not established that this gene acts upstream in the phenotypic pathway. In my opinion, without such analysis, the authors have just identified another interesting gene in the pathway, but not more.

2. I'm not convinced by the epistasis analysis of *seud-1*. It's not clear from the writing if null alleles of *nhx-40* were used. This seems key, as otherwise *seud-1* could still be acting either upstream or downstream.

Response to reviewers:

Reviewer #1:

I do think the Figures could be smartened up and in some cases designed to more effectively cover how the plasticity works and functions. Some issues here, or at least questions raised are: At which developmental stage are the nematodes sensitive to their environment – and especially to the presence/absence of potential prey? Also, how does this sensitivity relate to the pathways described in the figures and in particular to the working of the switch mechanism suggested in Fig 4. Why are there three dashed 'epigenetic' arrows in Fig 4 but only one for environment? Does this environmental influence of diet availability (or indeed those for epigenetics) coincide with the initiation or development of the pharyngeal morphology (apparently around J4-adult molt, Fig 2)?

Our response: We have substantially modified Figure 1 to provide a spatial context for explaining the *P. pacificus* polyphenism pathway as it is currently understood. Although the exact regulatory links between most steps in the pathway have yet to be worked out, it is known where some of the steps occur, at least in homologous processes in *C. elegans* (i.e., in dauer diapause). To explain the timing of the polyphenism switch to a more general audience, in the figure legend we reference previous studies of starvation sensitivity to environmental cues of starvation and pheromones. With regard to the epigenetic regulatory factors, too little is known about when they act in the animals, and we simply mention those regulators of *eud-1* for the sake of completeness, although we provide more explanation of them as well in the figure legend. The three arrows under the upstream regulators (epigenetics, pheromones, starvation) was simply an oversight in the figure design, which we have changed.

I note that in many other systems of polyphenism in insects the environmental cue can by some time precede hormone signals that in turn regulate differing expressions of developmental genes and pathways. Thus, even if details are elsewhere in the literature, I would like to see more clarity of the roles of 'starvation' and 'Ascarosides' at the top of Fig 1, as well as more clarity where possible (or if not possible some discussion thereof) of the top slice of Fig 4.

Our response: We now reference the timing of starvation and ascaroside cues in the legend of Fig. 1, we give the spatial context in an idealized nematode diagram we have added, and we refer to reviews of what is known about reception of these cues in *C. elegans*. By providing a much more detailed legend for Fig. 1, we hope that the simple annotations of Fig. 5 (previously Fig. 4), the legend of which we have also expanded, will provide more context for the general reader.

Personally, I would be more cautious about the 'optimizing morphology to the environment' phrase in Fig 1; matching or association perhaps but optimizing – have the necessary experiments been done, including in natural environments? In this line of thinking, the phrase at the start of para 2 in the introduction '... to take different diets in response to local environmental cues' could I think be rewritten to make it clearer what is meant.

Our response: We agree that “optimizing morphology” is not accurate, as the dimorphism essentially offers two options to respond to environmental cues. Therefore, we have reworded

the language (*i.e.*, to “matching”) in the figure legend and abstract to reflect this concern. Additionally, we have revised paragraph 2 of the introduction to be clearer about the plastic response of *P. pacificus* to starvation. Additionally, experiments assessing the relative fitness benefits of either morph on a prey diet have indeed been performed, and we now mention this and cite the appropriate reference, also in paragraph 2.

I would value some more information on sample sizes – for example in Fig 2 part C, how many individuals were assayed, how much variability was seen and also which stages of development were surveyed (similarly for S2; and text section: SEUD-1 is expressed at the site of the polyphenism)? Otherwise, I find the phrase ‘thus controlling the switch at the site of plasticity’ to be more open to doubt than the text in Fig legend suggests (again cf text para). This also goes back to uncertainty at least in this MS about the timings of the environmental sensitivity, the switch mechanism and the development itself.

Our response: Because information on sample sizes was not previously clear, an unfortunate error on our part, we are now explicit about these as well as the associated variation we found in *seud-1* expression. More substantially, we provide more detail on the ontogeny of expression, which we identify as taking place in dimorphic morphology-producing cells throughout postembryonic development. We have created a new figure (Fig. 3 in our resubmission) in the main text to show spatial expression at two life-stages, and we include expression images of all other postembryonic life stages, including the facultative dauer diapause larva, an environmentally induced alternative life-stage to direct-developing J3 larvae. We appreciate this suggestion, also made by Reviewer #2, to provide more detail in this regard, as this has enabled us to expand and be much more precise about the location, timing, and variation of expression.

In the Section headed Genetic inbreeding ..., the word ‘presumably’ is a red flag – what does it mean, how large was the voucher sample, what was limited and what were the culture conditions?

Our response: We were mistaken to use the word “presumably,” which does not do justice to the state of the field. The genome of *P. expectatus* has been completely sequenced and assembled, and in the genome publication (Rödelsperger *et al.* 2014, *Genetics*), levels of heterozygosity incurred during inbreeding were explicitly estimated; even after extreme inbreeding, *P. expectatus* was shown to harbor much more heterozygosity than *P. pacificus*. We have made this simple revision, and in the Methods we provide much more information on how the population was collected, maintained, and curated as a voucher.

I am not a nematode person, and so would like to know what are the likely fitness effects of ten generations of extreme (single pair) inbreeding on fitness; this seems extreme and would in the insects I work with a) be impossible and b) produce all sorts of negative effects on fitness and phenotype. What were the conditions of so-called artificial selection (for only two generations) and from what is detailed here I have no idea of how it was performed (e.g. intensity of selection, numbers used, assays performed and in which environmental conditions)? This relates, for example, to interpreting the final sentence of the Results. Perhaps some more details in a section in the SI.

Our response: We have provided more clarity on our inbreeding scheme, which was mostly lacking in our original submission. While in some nematode (*e.g.*, several *Caenorhabditis*) species, single-pair inbreeding for ten generations is likewise difficult or impossible, mostly unpublished results have shown that such inbreeding tends to be more feasible for gonochoristic *Pristionchus* species, and indeed published results have shown that it is feasible for *P. exspectatus*, which provided the motivation for us to perform our inbreeding experiment (as now mentioned in the text). However, achieving such an outcome is not trivial, as we now clarify in the Methods: inbreeding often does cause fitness decline, so high replication and storing breeding pairs from several previous generations are needed to inbreed the lines without observable fitness effects. In addition to clarifying the methods and rationale for this experiment, we also provide more detail on our simple artificial selection scheme.

In the introduction it is not clear to me what exactly the phrase ‘we identify and alternative polyphenism switch gene’ refers to given what goes before.

Our response: The language has been cleaned up.

--

Reviewer #2:

*While the discovery of *seud-1* and its analysis are very interesting and of high relevance, I have two main criticisms: First, the manuscript, data presentation and interpretation are often difficult to follow, and the writing is frequently vague or imprecise. Second, some of the experimental data are rather weak in supporting the conclusions drawn, in particular, concerning the evolution of *seud-1* and its link to the feeding morph polyphenism; a number of obvious, relatively simple experiments could be carried out to corroborate certain conclusions.*

I detail these two main points and additional comments below.

Our response: We agree that our language describing the phenotype, the genetics experiments, and our interpretation of those experiments could be improved. We have done so, with details following specific comments below.

*Starting with the abstract and introduction, I encountered many unclear statements and sentences regarding the study system. Often, generic terms, like plasticity or “plasticity phenotype” are used without making explicit reference to the phenotype studied (*e.g.* production of a specific morph type). The use of such generic terms introduces ambiguousness and confusion. For example: The statement “how genetic changes result in the evolution of novel plasticity phenotypes remain unknown” is unclear. What is a novel plasticity phenotype of interest here? Reaction norms? Morphology of alternative phenotypes? Similarly, the statement “Furthermore, we show that plasticity, by being sensitive to gene dosage, can evolve directly by gene amplification, in addition to expected stepwise transcriptional changes” is not explicit about what plasticity refers to.*

Our response: To remove ambiguousness and confusion from our description, we have revised text throughout the manuscript to be clear that the phenotype we are studying is the morph-ratio,

or the plastic response of a dimorphism to environmental cues.

For readers not very familiar with the study system, I would guess there is some confusion about the definition of plasticity and environmental cues regulating the induction of phenotypes. In classical cases of plasticity/polyphenism, there are specific cues inducing alternative phenotypes; however, in the case studied here, alternative phenotypes are generated even under a constant environment (e.g. bacteria only). The relative production of the alternative phenotypes is thus not tightly linked to the presence of alternatives cues – perhaps this should be made clear; moreover, none of the experimental work involves any modulation of environmental cues. It therefore remains unclear, how environmental variation connects to the production of alternative feeding morphs and how this relates to the genetics studied. I would therefore recommend explaining the study system better in the introduction, in particular, to elaborate on how morph allocation occurs in constant environment, and why natural isolates produce 90% Eu morphs in bacterial environments.

Our response: We appreciate this suggestion, and we have now modified our introduction to make the details of our model system more accessible to the general reader. Indeed, the laboratory reference strain of the nematode *P. pacificus* is less developmentally “plastic” than (*i.e.*, the relative production of alternative phenotypes is not as tightly linked as in) other strains and particularly other species, at least as studied in the lab, which we now explain and reference in the introduction. This interesting feature of *Pristionchus* as a model for developmental plasticity allows us to study variation in standing morph-ratio responses to standardized environmental cues, a feature we have been able to harness here and previously. Moreover, we are also thankful for the suggestion, also made by Reviewer #3, to examine the effects of environmental cues on the polyphenism switch, which we have now done. Specifically, we have performed a comparative RNA-Seq analysis on strains exposed to food sources other than laboratory OP50 and which are known to signal different morph-ratio responses in *P. pacificus*, the results of which we present in Table 1.

RESULTS / DATA

- Data on morph production of recovered mutants is not quantitatively shown – what are their effect sizes, expressivities? Are they sensitive to environmental variation?

Our response: We acknowledge that we were unduly terse with our presentation of our mutant phenotype data, and we did not even report quantified phenotypes of the suppressor mutants. We apologize for this unnecessary omission. We have rectified this, adding a new subfigure (Fig. 2A) to show the phenotypes of mutants and suppressors. Furthermore, we are explicit that, even in a St-promoting *eud-1* mutant background, the null mutations are fully penetrant, regardless of a relatively “St-promoting” environment – this environment induces about 90% Eu in the wild-type and 0% in *eud-1* mutants. Moreover, no change was expected or observed under starved or crowding conditions, which would drive this value higher, because *seud-1* mutants already have a 100% phenotype.

- Expression analysis using the reporter strain is minimalistic - what is the time course of

*expression? How does expression differ between different morphs? Can environmental cues modulate *seud-1* expression?*

Our response: We have greatly expanded our spatial expression analysis to show expression at all postembryonic life-stages, establishing two new figures (Fig. 3 and Fig. S2), and we have consequently rewritten our results of this study to provide much more detail. Whether expression differs in different morphs presents a quandary, because the sensitivity and activity of the switch occurs before the morphs can be identified (only the adults are dimorphic). Indeed, we collected expression data for St adults (one is shown in Fig. 3) and Eu adults, although expression was weaker in adults in general (as would be expected for a developmental decision) and did not reveal qualitative differences. To get around this dilemma, and to provide a way of quantifying differences, we analyzed transcript counts under different morph-inducing environments, so that transcripts included those produced by juveniles before the polyphenism decision was irreversibly made (Table 1). As a result, we found that environmental cues do in fact modulate *seud-1* expression, which we report.

- Epistasis analysis is primarily based on over-expression lines – are there are no relevant null alleles?

Our response: We acknowledge that we could provide more data and explanation of the epistasis analysis. Because loss-of-function (*lof*) mutations in both *seud-1* and *nhr-40* (both of which are suppressor-of Eud mutants) results in an all-Eu phenotype, whereas wild-type alleles of both genes promote the St morph, it was not possible in principle to perform epistasis tests with double *lof* mutants. We have clarified the design of our test in the text, and we now also include phenotype data on both *lof* alleles and over-expression lines for both genes, as well as the epistasis tests between them (Fig. 3C). Also, we clarify that our *seud-1* mutation used in these tests was indeed a null allele, and the *nhr-40* mutant a fully penetrant *lof* mutant for its polyphenism (*i.e.*, St-promoting) phenotype (Kieninger *et al.* 2016, *Curr. Biol.*; more detail given in response to Reviewer #3 below).

*- There is no analysis of potential variation in *seud-1* (sequence/expression, e.g. of transcripts) across different *P. pacificus* wild isolates and connection to production of alternative phenotypes. This would be rather obvious and relevant to do.*

Our response: This was a good suggestion, and consequently we performed RT-qPCR of *seud-1* and *eud-1* in three divergent *P. pacificus* strains held at standardized lab conditions. As a result, we report significant differences in the expression of both genes (Fig. S3). In particular, we found that *eud-1* was lower in St-biased strains, as expected, although *seud-1* was also downregulated in these strains, suggesting that more than just the ratio of these two genes is necessary to explain the observed variation. This result was not very surprising given the genetic divergence of these particular strains (Rödelsperger *et al.* 2014, *Genetics*), and we are consequently conservative in how we interpret these results in the text. Because of the variable of long haplotype distances involved in this experiment, we argue that our original test to select for differences from a single starting population (Fig. 4C-4E) is more appropriate to immediately capture how changes in expression can correlate with phenotypic differences on a more immediate time scale.

- *Genetics: The genetic analyses presented in Figures 3A and 3C are somewhat difficult to follow (the labelling of crosses could be simplified for figures). This is further complicated by the rather non-standard genetic terms used to describe results of crosses, and subsequent interpretations. For example, I don't see the need to talk about "competition" among the two enzymes and the sentence "Because a plasticity is set by the competition of two opposing enzymes, selection for differential regulation or genomic amplification of these factors provides a binary "dial" for a plasticity to evolve" is unnecessarily difficult.*

Our response: Because we agree that the language of our genetics narrative was imprecise in places, we have rewritten our manuscript to be more accurate about the phenomena we describe, namely in terms of dosage (not "dial"), epistasis, and – where relevant – opposing (not "competitive") function of the sulfatase and sulfotransferase, a known phenomenon, as described in papers we cite in our Discussion. We have rewritten several of our interpretations of the genetics results, including the one pointed out here.

In addition, despite some explanations in the materials and methods section, I didn't fully understand how statistics were performed (e.g. frequency comparisons pooling of replicate crosses?).

Our response: We have performed a more appropriate statistical analysis (*i.e.*, using generalized linear models), and we provide details about this when reporting our test statistics and in the Methods. While we did not expect our test results to qualitatively change, we acknowledge that increased rigor and transparency of our statistics have improved the manuscript.

- *Analysis of *seud-1* duplicates in *P. exspectatus*: I found the hybrid genetics involving the two species rather weak and indirect experimental evidence to make the argument that "Gene amplification results in new plasticity phenotypes across species". Why not simply knock out/down these genes in this species?*

Our response: While gene amplification in *P. exspectatus* is likely, given the weight of evidence, we agree that our interpretation is not conclusive, so we have rephrased our statements asserting cause and effect. Although we maintain that our hybrid cross results clearly support our inference of differing gene-copy ratios (*i.e.* double "heterozygotes" vs. double "homozygotes") correlating with the resultant morph-ratio phenotypes, we acknowledge that a functional test, as is possible in *P. pacificus*, would be the "gold standard" for this kind of inference. Unfortunately, gene knockdown (*i.e.*, RNAi) has, in spite of multiple published attempts, proven to be infeasible in tested *Pristionchus* nematodes, and CRISPR/Cas9 has likewise not yet worked in non-*P. pacificus* species of *Pristionchus*, at least to our knowledge. While my lab is currently pioneering CRISPR/Cas9 strategies on other nematodes, including non-hermaphroditic (*i.e.*, genetically more intractable) diplogastrid species such as *P. exspectatus*, we feel that this line of experimentation and the technical advance it requires to be outside the scope of this manuscript. Consequently, we have rewritten our manuscript to more conservatively assert that there is a definite correlation, but not conclusive causation, between copy number and phenotype, and we reserve our statements on "gene amplification" as speculation in the Discussion.

- The experiments on “Laboratory evolution of plasticity occurs through competing switch genes” involved derivation of two lines by some initial selection plus inbreeding are in my view not very conclusive. While suggestive, they do not causally demonstrate that difference in *Eu morph* production stems from measured *seud-1* differences.

Our response: While we maintain that the correlation of expression with phenotype is strong (i.e., suggestive, as the reviewer points out), we agree that the evidence is not definitively causal and have rephrased our interpretations accordingly.

- Given the above points, I think multiple conclusions/statements in the paper are premature, including the following:

- “Abstract: Further, recent duplication and laboratory selection of *seud-1* in a sibling species has amplified the gene’s control of the polyphenism’s threshold, identifying this gene as a target for plasticity evolution.”

- “Taken together, these results show that opposing switch factors have been amplified and are experiencing regulatory divergence in this species, resulting in a new plasticity response to a given environmental regimen.”

- In conclusion, we have identified a genetic mechanism for establishing a threshold between plastic developmental phenotypes, and we have shown how – in the case of *P. expectatus* – such a mechanism has been modified to produce a novel plasticity response.

Our response: In addition to supplying multiple new experiments to support our conclusions of the MS, as described above, we have rewritten these statements (highlighted in red text in the manuscript) to more accurately summarize our results.

- In addition, certain claims are somewhat exaggerated, e.g. in the abstract “Our findings thus provide the first genetic insight into how the molecular regulation of polyphenism evolves to produce new plasticity phenotypes“. This statement is incorrect – a number of examples in insect polyphenisms or *C. elegans* dauer formation are known.

Our response: It is unfortunate we gave the impression of glossing over extensive research on the genetics or evolution of plasticity – indeed, I was recently invited to write a review on exactly the topic of polyphenism genetics and evolution, a field of fast-growing interest (Projecto-Garcia *et al.* 2017, *Curr. Opin. Genet. Dev.*). From how we understand the field, we stand by our assertion that our combination of genetic and evolutionary approaches to polyphenism research is unprecedented, given the unique macroevolutionary experiments and analyses we present in our study. However, we also do not want to make any misleading exaggeration, so we have simply eliminated the word “first” from this statement.

--

Reviewer #3:

*The paper is clearly written, and the experiments are appropriate. The use of a “non-standard” model system is exciting, although by this point *P. pacificus* has many tools available. I have two*

main issues that I think are important to address that may impact the significance of the findings:

Our response: We have revised how we label our model system.

1. The authors set up the paper as a study to understand the origin of polyphenism. However, they appear to fall short of this. The problem is that they look at mutants or isolates having increased or reduced expression of genes and show that phenotypic ratios can be altered. But that does not indicate that the gene involved is the actual site of regulation when phenotypic choices are made. All that could be happening here is that they are altering the set points for the system, but not the regulatory points. It seems to me that the only way to convincingly demonstrate that a gene is the site of phenotypic control, is to track its expression or activity in animals displaying the different phenotypes and show a 1-1 correlation. Even that may not be sufficient if it is not established that this gene acts upstream in the phenotypic pathway. In my opinion, without such analysis, the authors have just identified another interesting gene in the pathway, but not more.

Our response: The reviewer raises an important point about the need to demonstrate environmental control of the switch mechanism. In response to this excellent suggestion (mentioned also by Reviewer #2), we have now demonstrated this (Table 1). Further evidence that argues *seud-1* is a part of the switch mechanism, and not simply a “stepwise function” gene (or any other gene in a battery of polyphenism genes) following the actual decision, is that *seud-1* acts upstream of a nuclear hormone receptor (*nhr-40*) in the polyphenism. Of course, as the reviewer points out, polarizing steps in the pathway depends on proper epistasis analysis, an issue I address below. Additionally, it may be worth clarifying here that *seud-1*, like both other *P. pacificus* polyphenism “switch genes” described to date, only affect the morph ratio, and not an aspect of the dimorphic morphology being specified (*i.e.*, some phenotype downstream of the polyphenism decision).

2. I'm not convinced by the epistasis analysis of *seud-1*. It's not clear from the writing in null alleles of *nhr-40* were used. This seems key, as otherwise *seud-1* could still be acting either upstream or downstream.

Our response: We now explain in the text that we use a mutant with a complete loss-of-function allele, but not a null (*i.e.*, loss of gene product) mutant, which does not exist for *nhr-40* in *P. pacificus*. To clarify this point, as we have clarified in the text, all previously recovered *nhr-40* mutants were determined to be *lof* for the polyphenism (St-promoting) phenotype (Kieninger *et al.* 2016, *Curr. Biol.*), even if not *lof* for all potential phenotypes the gene influences, which would be expected from a loss of gene product. Presumably, given the pleiotropy of *nhr-40*, null alleles of *nhr-40* would produce a very harmful or lethal phenotype, as described for a *C. elegans* *nhr-40* null allele (Brozova *et al.* 2006, *Mech. Dev.*). Thus, we assert that the *completely penetrant* (all-Eu, and suppressor-of-Eud) phenotype of our *nhr-40* mutant indicates that it is not simply a reduction-of-function mutation and is appropriate for our epistasis tests. We kindly point out that better-characterized NHRs in *C. elegans* have mutations that putatively eliminate their interactions with other proteins, which in principle could include those co-regulators specific to a certain phenotype, thus allowing the mutant NHR to retain its function for other

processes. Thus, we hope that our revision clarifies our experimental design and alleviates this concern of the reviewer.

Reviewers' Comments:

Reviewer #1:

Remarks to the Author:

This is a much improved manuscript (including the figures) which from my perspective should be published because it makes an important contribution to understanding the evolution of phenotypic plasticity especially at a molecular level.

In line 59, 'a diet of only prey' requires rewording.

In the final para on p 8 (and see p 15) before the Discussion I would argue that the experiment here does not involve what I as a quantitative geneticist working with insects would call artificial selection. Rather it is producing inbred lines and the fact that these differ in plasticity phenotype (morph ratio) does indeed imply that there is genetic variation among inbred lines (and in the original population). Artificial selection, however, involves in my view selecting for particular phenotypes as parents in a series of generations to select populations in different directions or trajectories in phenotypic space - responses indicate genetic variation contributing to the phenotypic variation. To me this is an important distinction but I am not saying that the interpretation of the inbred line experiment is wrong. I just don't think that the term artificial selection should be used in this context (as I now understand the experimental protocol better).

Reviewer #2:

Remarks to the Author:

The authors have aimed to address most points raised by the different reviews, and the revised version is substantially improved. Some of the experimental evidence remains rather weak, however, the authors have been more careful in the interpretation of these results.

One question that I had before remains: Is there any natural DNA sequence variation in *seud-1* that could (potentially) explain variation in morph ratio? Any (genetic/molecular) hypothesis to why expression levels of the three isolates differ?

Reviewer #3:

Remarks to the Author:

The revised manuscript from the Ragsdale lab is significantly improved over the initial submission. In particular, the demonstration of level changes of *seud-1* under different conditions is quite convincing, and makes it likely this gene is a major point of regulatory control for phenotype switching.

I still take issue, however, with the epistasis analysis. The authors state in the paper that "*seud-1* either acts downstream of *NHR-40* or, alternatively, is necessary to sulfate a signal that ultimately activates *NHR-40*-mediated transcription. In the former model, *seud-1* over expression should suppress the phenotype of *nhr-40* mutants, whereas in the latter model the reverse should be the case."

Since the *nhr-40* allele is not a null, overexpression of *seud-1* could either bypass *nhr-40*, thereby acting downstream, or drive activation of the less active *nhr-40* by sulfating it at a higher level than the wild-type to provide sufficient activity, therefore acting upstream. This experiment therefore is not sufficient to test epistasis. Overexpressing *nhr-40* in the *seud-1* null allele, however, should, however, give a much stronger answer. Have the authors tried this?

In any case, the epistasis studies are not absolutely crucial to the main point of the paper, and so I would be comfortable with the authors just discussing the alternatives in a more cautious tone.

Response to reviewers' comments

Reviewer #1 (Remarks to the Author):

In line 59, 'a diet of only prey' requires rewording.

Our response: we have reworded this sentence to "...deprived of bacteria and offered nematodes as prey."

In the final para on p 8 (and see p 15) before the Discussion I would argue that the experiment here does not involve what I as a quantitative geneticist working with insects would call artificial selection. Rather it is producing inbred lines and the fact that these differ in plasticity phenotype (morph ratio) does indeed imply that there is genetic variation among inbred lines (and in the original population). Artificial selection, however, involves in my view selecting for particular phenotypes as parents in a series of generations to select populations in different directions or trajectories in phenotypic space - responses indicate genetic variation contributing to the phenotypic variation. To me this is an important distinction but I am not saying that the interpretation of the inbred line experiment is wrong. I just don't think that the term artificial selection should be used in this context (as I now understand the experimental protocol better).

Our response: we have eliminated references to this experiment as "artificial selection" and instead refer to it as "systematic inbreeding," which more accurately captures describes the experiment.

--

Reviewer #2 (Remarks to the Author):

The authors have aimed to address most points raised by the different reviews, and the revised version is substantially improved. Some of the experimental evidence remains rather weak, however, the authors have been more careful in the interpretation of these results.

*One question that I had before remains: Is there any natural DNA sequence variation in *seud-1* that could (potentially) explain variation in morph ratio? Any (genetic/molecular) hypothesis to why expression levels of the three isolates differ?*

Our response: we have added a sentence to the text explaining that no explanatory sequence differences could be identified, and we briefly offer hypotheses for what may in fact be different among phenotypically divergent lines.

--

Reviewer #3 (Remarks to the Author):

*I still take issue, however, with the epistasis analysis. The authors state in the paper that "*seud-1* either acts downstream of *NHR-40* or, alternatively, is necessary to sulfate a signal that*

ultimately activates NHR-40-mediated transcription. In the former model, seud-1 over expression should suppress the phenotype of nhr-40 mutants, whereas in the latter model the reverse should be the case."

Since the nhr-40 allele is not a null, overexpression of seud-1 could either bypass nhr-40, thereby acting downstream, or drive activation of the less active nhr-40 by sulfating it at a higher level than the wild-type to provide sufficient activity, therefore acting upstream. This experiment therefore is not sufficient to test epistasis. Overexpressing nhr-40 in the seud-1 null allele, however, should, however, give a much stronger answer. Have the authors tried this?

Our response: We agree that the over-expression of *nhr-40* in a null *seud-1* background is necessary for proper epistasis analysis. We did in fact include the results of exactly this experiment in our submission (Fig. 2C, last bar), although we recognize that the explanatory logic and presentation of our results was unnecessarily terse. We have therefore provided more detail in this paragraph of the text. Also, we kindly point out that *seud-1* over-expression did not change the phenotype of the *nhr-40* mutant, and so we also provide more detail explaining our results in the text.

In any case, the epistasis studies are not absolutely crucial to the main point of the paper, and so I would be comfortable with the authors just discussing the alternatives in a more cautious tone.

Our response: We hope that by drawing clearer attention to the results of the above-mentioned experiment, that our interpretation adequately represents our results.